# Beyond Multiparametric MRI and towards Radiomics to Detect Prostate Cancer: A Machine Learning Model to Predict Clinically Significant Lesions

**DOI:** 10.3390/cancers14246156

**Published:** 2022-12-14

**Authors:** Caterina Gaudiano, Margherita Mottola, Lorenzo Bianchi, Beniamino Corcioni, Arrigo Cattabriga, Maria Adriana Cocozza, Antonino Palmeri, Francesca Coppola, Francesca Giunchi, Riccardo Schiavina, Michelangelo Fiorentino, Eugenio Brunocilla, Rita Golfieri, Alessandro Bevilacqua

**Affiliations:** 1Department of Radiology, IRCCS Azienda Ospedaliero-Universitaria di Bologna, 40138 Bologna, Italy; 2Department of Experimental, Diagnostic and Specialty Medicine (DIMES), University of Bologna, 40138 Bologna, Italy; 3Division of Urology, IRCCS Azienda Ospedaliero-Universitaria di Bologna, 40138 Bologna, Italy; 4Radiology Unit, “Infermi” Hospital, 48018 Faenza, Italy; 5SIRM Foundation, Italian Society of Medical and Interventional Radiology, 20122 Milano, Italy; 6Department of Pathology, IRCCS Azienda Ospedaliero-Universitaria di Bologna, 40138 Bologna, Italy; 7Department of Computer Science and Engineering (DISI), University of Bologna, 40126 Bologna, Italy

**Keywords:** cancer staging, machine learning, multiparametric magnetic resonance imaging, prostate cancer, radiomics

## Abstract

**Simple Summary:**

Early diagnosing clinically significant prostate cancer (csPCa) through Magnetic Resonance Imaging (MRI) is very challenging and, nowadays, csPCa confirmation comes exclusively from prostate biopsy. However, biopsy is an invasive procedure and it also frequently causes csPCa misclassification. This study develops a non-invasive machine learning method for early predicting csPCa based on radiomic MRI image analysis. The main novelty of this study is investigating the radiomic differences between progressive risk groups of PCa, attributed according to the biopsy outcome and Gleason Grade group stratification. Besides predicting csPCa with very good performance (sensitivity = specificity = 84% in the test phase), this study highlights patients with GG = 2 as non-csPCa (being statistically equivalent to GG = 1), so that GG = 2 can be counselled for follow up, whilst GG ≥ 3 admitted to radical treatments. Not least, this study provides a plausible clinical interpretation of radiomic features, by discussing their values with respect to the histological meaning.

**Abstract:**

The risk of misclassifying clinically significant prostate cancer (csPCa) by multiparametric magnetic resonance imaging is consistent, also using the updated PIRADS score and although different definitions of csPCa, patients with Gleason Grade group (GG) ≥ 3 have a significantly worse prognosis. This study aims to develop a machine learning model predicting csPCa (i.e., any GG ≥ 3 lesion at target biopsy) by mpMRI radiomic features and analyzing similarities between GG groups. One hundred and two patients with 117 PIRADS ≥ 3 lesions at mpMRI underwent target+systematic biopsy, providing histologic diagnosis of PCa, 61 GG < 3 and 56 GG ≥ 3. Features were generated locally from an apparent diffusion coefficient and selected, using the LASSO method and Wilcoxon rank-sum test (*p* < 0.001), to achieve only four features. After data augmentation, the features were exploited to train a support vector machine classifier, subsequently validated on a test set. To assess the results, Kruskal–Wallis and Wilcoxon rank-sum tests (*p* < 0.001) and receiver operating characteristic (ROC)-related metrics were used. GG1 and GG2 were equivalent (*p* = 0.26), whilst clear separations between either GG[1,2] and GG ≥ 3 exist (*p* < 10${}^{-6}$). On the test set, the area under the curve = 0.88 (95% CI, 0.68–0.94), with positive and negative predictive values being 84%. The features retain a histological interpretation. Our model hints at GG2 being much more similar to GG1 than GG ≥ 3.

## 1. Introduction

In the last decade, the strategy used to diagnose prostate cancer (PCa) in patients with a clinical suspicion of PCa has radically changed from systematic biopsy to the routine use of multiparametric magnetic resonance imaging (mpMRI) in the first instance. The so-called “MRI-pathway” [1] has dramatically changed clinical practice since, in many health systems, mpMRI could be used as a triage test to avoid a biopsy if the results were negative [2], whereas positive results could be used for targeting suspicious lesions during the prostatic biopsy. Thus, mpMRI-targeted biopsy has proven to increase the diagnosis of clinically significant PCa (csPCa) compared to systematic biopsy alone [1]. However, a recent systematic review found that mpMRI had a wide range of diagnostic accuracy with a sensitivity (SN) of 58–96%, negative predictive value (NPV) of 63–98%, and specificity (SP) between 23% and 87% [3]. In this regard, while the low SP may be overcome by subsequent systematic biopsy, the false positive mpMRI lesion obscures the diagnostic process and prompts unnecessary biopsies [2]. In fact, despite numerous efforts to standardize settings and reports [4], mpMRI invariably has limitations, including variability in scan quality or reporting [3], moderate inter-reader variability [5] and a steep learning curve.

To note, discerning csPCa from benign processes or non-csPCa is crucial since patients’ management may vary dramatically. Different definitions are reported to identify csPCa [6]. However, patients with Gleason Grade group (GG) ≥ 3 have a significantly worse prognosis compared to ones with GG1 and 2 [7]. Thus the identification of men with such risk of unfavorable oncologic outcomes is essential to offer best treatments. Even in the best scenario, the risk of misclassification of PCa by mpMRI parameters is consistent, also using the updated prostate imaging reporting and data system (PIRADS) score, which, although related to the increased risk for csPCa, is not directly associated with GGs. Several MRI-derived parameters were investigated to identify high-risk features in pathology. For example, the lower the apparent diffusion coefficient (ADC) value, the higher risk of the cribriform pattern is reported [8]. However, mpMRI-derived parameters are mainly visual features and, as such, are limited by what is perceivable even by the most expert human eye.

Recently, radiomics has emerged as the application of well-established machine learning (ML) and artificial intelligence techniques to medical image analysis [9,10,11]. The large popularity gained by radiomics arises from the possibility of ML algorithms to considerably enrich the information retrievable from imaging, represented by latent radiomic features, which ultimately allow improvements in the diagnostic potential of mpMRI [12,13].

Accordingly, the main goal of our research is to develop a ML model to predict csPCa, namely, GG ≥ 3. In particular, this study sets up an image processing procedure based on extracting local radiomic features, enabling the visual assessment of colorimetric maps on the whole prostate gland, through which similarities and differences between GG1, GG2 and GG ≥ 3 are analyzed. This allows the prediction of high-risk diseases even before any biopsy examination.

## 2. Materials and Methods

### 2.1. Patient Cohort

This study was observational, retrospective and single-centered. It was approved by our local Institutional Review Board (IRB), which waived the need for informed consent, and it was conducted in accordance with institutional guidelines, including the Declaration of Helsinki (Ethics Committee code: 784/2021/Oss/AOUBo). Two hundred and twenty patients with at least one PIRADS ≥ 3 lesion at mpMRI, according to the European Society of Urogenital Radiology (ESUR) guidelines v2.1, performed at our Radiology Unit from September 2020 to December 2021, were enrolled. The inclusion criteria were the following: (1) having undergone fusion-TB of the index lesion at our Radiology Unit and (2) having a diagnosis of PCa at a histopathological report of TB from a dedicated pathologist of the Pathology Unit of our Institution. The exclusion criteria were the following: (1) mpMRI protocol not completely adhering to the suggested imaging protocols described in the ESUR guidelines; (2) previous prostatic surgery or hormone therapy; (3) the presence of severe artifacts not allowing the evaluation of one or more sequences of the mpMRI due to uni/bilateral hip prostheses or other causes. Finally, 102 patients are analyzed, and the flowchart of patient enrolment is reported in Figure 1.

### 2.2. Image Acquisition

The mpMRI examinations were carried out using a 1.5 T scanner (Signa HDxt; GE Healthcare, Chicago, IL, USA) and a pelvic phased-array surface coil combined with a disposable endorectal coil. Although the use of the 3 T scanner is recommended by many authors, the ESUR guidelines also specify that an optimization of the acquisition parameters and adequate technical requirements provide adequate and reliable diagnostic examinations even with the 1.5 T scanner. Therefore, the mpMRI study of the prostate gland and seminal vesicles is optimized according to the ESUR guidelines and includs Fast Relaxation Fast Spin Echo (FR-FSE) T2w, DWI and Dynamic Contrast-Enhanced (DCE) sequences; the scan parameters were described in a previous study [14]. Appendix A reports the scan parameter of the DWI, processed to obtain ADC maps using the couple b = 50, 1000 s/mm${}^{2}$.

### 2.3. Prostate Biopsy and Pathological Analysis

All PIRADS ≥ 3 lesions underwent target biopsy performed transrectally by two experienced radiologists using the mpMRI-TRUS fusion image guide after antibiotic prophylaxis and a cleansing rectal enema using a non-disposable biopsy gun (Medgun, Medax, Modena, Italy) with a disposable 18-gauge needle and a US platform (Canon-Toshiba Aplio 500TM, Ōtawara, Kanto, Japan) with an end-fire TRUS probe as previously described [15]. The biopsy was performed 32.2 ± 11.4 days after mpMRI, and samples were analyzed by two genitourinary pathologists who graded neoplastic lesions from 1 to 5, according to GG. Finally, 117 PCa lesions were confirmed, 61 GG < 3 and 56 GG ≥ 3.

### 2.4. Regions of Interest (ROIs) Outlining

In subsequent radiomic analysis, we decided for exclusively ADC inclusion due to its role in the evaluation of both peripheral and transition zones; this also favors the suitability of our analysis in biparametric MRI protocols. Then, after the mpMRI studies were retrieved from the picture archiving and communication system, manual segmentation was performed on ADC, checking corresponding findings on T2. Both the suspicious nodules and the whole prostate gland were outlined, following their visible borders. After proper windowing, each radiologist contoured the regions of interest (ROIs) of those patients on which he had previously performed fusion biopsy, using ImageJ v1.53 (https://imagej.nih.gov/ij, accessed in 13 August 2020), a Java-based public-domain software [16].

### 2.5. Radiomic Feature Generation

Altogether, 132 radiomic features were generated from ADC, exploiting the local approach proposed in [12], already applied in [17,18], and shown in Figure 2 (using representative metrics).

In particular, 10 first-order features (i.e., mean, median, kurtosis, skewness, entropy, uniformity, interquartile range, coefficient of variation, standard deviation and median absolute deviation) were computed locally within the prostate ROI (Figure 2a) where each pixel is assigned a first-order feature calculated on a rectangular window (centered on the pixel itself) (Figure 2b). Thus, 10 (colorimetric) parametric maps of the features were achieved and filtered by the binary masks of the PCa ROIs (Figure 2c). After preliminary tests, a window of 9 × 9 pixel size was chosen as the most appropriate one. Then, the pixel-based measurements within the PCa were summarized in single values through 12 global descriptors (i.e., the same 10 first-order features as above, plus the mean and median of the last decile of the pixel distribution) (Figure 2d); this yielded 120 radiomic features (Figure 2e). In addition, the 12 global descriptors were also computed on ADC maps (Figure 2f).

### 2.6. Radiomic Feature Selection

Based on the smallest sample-class size of the study population (i.e., 56 GG ≥ 3), we decided to include at most four radiomic features to build the “radiomic signature” predictive of GG ≥ 3. Starting from the 132 features, feature selection was performed through a two-stage procedure carried out after a linear min-max feature normalization. First, a candidate feature subset was searched for through the least absolute shrinkage and selection operator (LASSO) regression by exploiting 10-fold cross-validation (CV) at the minimum CV error rule, and weighing each sample by the prior probability of its membership class (i.e., GG < 3 and GG ≥ 3). Then, the final feature vector was searched extensively by training as many support vector machines (SVMs) (with linear kernel) as the total number of all possible combinations of four features coming from the first stage. The final combination was selected as the one having the highest informedness (I) among the most discriminant ones, that is, with the smallest *p*-value at Wilcoxon rank-sum test, *p* < 10${}^{-3}$, also considering Holm–Bonferroni correction.

### 2.7. Similarity Analysis among GGs

A similarity analysis among different GGs was carried out using the 4-dim feature vector selected at the previous stage in order to confirm the clinical hypothesis of this research on our study population. First, the Kruskal–Wallis (*p* < 10${}^{-3}$) test was exploited for multiple comparisons among GG = 1, GG = 2 and GG = 3 and extended to GG ≥ 3. Meanwhile, the one-tail Wilcoxon rank-sum (*p* < 10${}^{-3}$) test was used for pairwise comparison to verify whether there was a higher similarity between GG = 1 and GG = 2 than between GG = 2 and GG = 3 or GG ≥ 3. In addition, GG ≥ 3 groups were also tested for multiple differences (*p* < 10${}^{-3}$) and, if any, pairwise differences were tested as well. The similarity analysis was also visually carried out through boxplots. Finally, single-group variances were assessed and compared among GGs through the Ansari–Bradley test, performed after median removal (*p* < 0.05).

### 2.8. Training and Test of the Predictive Model of GG ≥ 3

First, the four selected features were augmented according to the procedure in [18] to increase the statistical significance of both training and test sets. The oversampled features had 200 samples each, equally split between GG < 3 and GG ≥ 3, namely the negative (N) and positive (P) classes, respectively. To assess the statistical representativeness of augmented features, a preliminary model was developed on the original population and compared to the ultimate classifier, hereby described (details in Appendix A). A linear SVM classifier was adopted for predictive model development because it can work well even with reduced sample datasets, being based on support vector (SV) placement within the feature space. In addition, the linear kernel uses the minimum number of hyperparameters, thus contributing to minimizing the risk of overfitting the original population. Finally, this choice allowed us to perform a fair comparison of the models developed on the original population and the augmented ones. As regards the definite model developed on the oversampled dataset, three-quarters of the total samples were assigned to the training phase, using the SVM margin rule [12,18], whilst the last quarter has been left for the external test by keeping an equal proportion of P and N instances in each subset. The SVM training employed 100 runs of 3-fold CV, and for each run, the SVM hyperparameters, linear scale γ and misclassification cost *C* have been estimated using the built-in MatLab Bayesian optimization algorithm [19].

For each trained SVM, the likelihood that a sample belongs to the estimated class, namely, the radiomic score, was predicted using a binomial logit function. A receiver operating characteristics (ROC) curve was built, and its area under the curve (AUC) was considered to discard all trained models most prone to overfitting; that is, those having AUC higher on the test set than on the training ones. For each SVM run, the model reaching the highest AUC on the test set was selected. Finally, the remaining models were assessed on the entire training set and sorted based on AUC and I descending values. The classifier with the highest performance of both metrics was selected and validated on the holdout test set. To assess classifier performance, AUC, SN, SP, I, positive predictive value (PPV) and NPV were used.

## 3. Results

### 3.1. Patient and PCa Lesions Characteristics

The clinical characteristics of the patient cohort are summarized in Table 1, whilst Table 2 reports lesion data.

Globally, the segmented PCa ROIs range within [5, 1007] mm${}^{2}$ (i.e., [8, 1655] pixels), with mean, median and IQR values equal to 97 (i.e., 159 pixels), 63 (i.e., 104 pixels) and 76 (i.e., 125 pixels) mm${}^{2}$, respectively.

### 3.2. Selected Features

Starting from 132 features, LASSO detected a preliminary subset of 17 features. Finally, 88 4-dim feature vectors’ results were significant (*p* < 10${}^{-7}$), the most discriminant of which was selected with I=0.68. This ultimate feature vector was made by the median of the local coefficient of variation (cv−m), the uniformity of the local mean (μ−u), the skewness of the local skewness (s−s) and the IQR referred to the standard deviation (σ−IQR). Figure 3 shows an example of a representative ADC slice (a) of a non-csPCa (GG2) of the four colorimetric maps of cv (b), μ (c), *s* (d) and σ (e).

### 3.3. Similarity between GG Groups

The discrimination between GG = 1, GG = 2 and GG ≥ 3 achieved by the radiomic signature are shown in the boxplot in Figure 4a, and significant differences exist among all groups (*p*∼10${}^{-11}$).

No significant difference between GG = 1 and GG = 2 (*p* = 0.26) is reported, whilst clear separations between either GG = 1 or GG = 2 and GG = 3 exist (*p*∼10${}^{-6}$). Even stronger separations result between GG = 1 and GG ≥ 3, and GG = 2 and GG ≥ 3, respectively (*p* < 10${}^{-8}$), whilst among GG ≥ 3 groups *p* = 0.94. The medians of GG = 1, 2, ≥ 3 progressively increase; that is −1.19, −0.98 and 0.99, respectively, whilst their variances remain comparable (*p* > 0.05), with $\sigma_{\left(GG=1\right)}^{2}$ = 2.87, $\sigma_{\left(GG=2\right)}^{2}$ = 2.46 and $\sigma_{\left(GG=3\right)}^{2}$ = 2.55.

### 3.4. Prediction of GG ≥ 3

The developed SVM predictive model had γ=0.0366 and C=0.0010. The radiomic signature is reported in Equation (Equation 1), where, in decreasing order, the features cv−m, σ−IQR, μ−u and s−s hold the greatest importance.
(1)g(x)=−0.04+1.67·cv−m+0.96·μ−u+0.22·s−s−1.12·σ−IQR

While features cv−m, μ−u and s-s, all have positive weights, we assume higher values for GG ≥ 3 and lower ones for GG < 3, σ−IQR shows an opposite behavior. Table 3 reports the performance evaluation metrics of the signature for the training and test sets separately, whilst Figure 3b,c shows the corresponding ROC curves.

In the training set (Figure 4b), AUC = 0.90 (95% CI, 0.84–0.95), with SN = 85% and SP = 87%, this yields I = 0.72. In the test set (Figure 4c), AUC = 0.88 (95% CI, 0.68–0.94), with both SN = SP = 84%, which corresponds to I = 0.68. Furthermore, the PPV and NPV results are 86% and 85% in the training set, respectively, whilst they are both equal to 84% in the test set. Hence, the prediction of GG ≥ 3 is achieved with 10 FP and 11 FN in the training set and 4 FP and FN errors in the test set. In this regard, Figure 3 also reports the waterfall plots for the training (Figure 4d) and test (Figure 4e) sets. For the sake of completeness, the Appendix A report the performance of the preliminary model developed without any feature sample augmentation.

## 4. Discussion

In the clinical management of organ-confined PCa, active surveillance is a safe alternative to radical treatments for a low-risk disease (GG1) [20], whilst its endorsement for favorable-intermediate risk PCa (GG = 2) is still debated [21,22]. Moreover, novel focal ablative treatments are attractive options for low/intermediate-risk disease, aiming at sparing treatment of healthy prostatic glands and reducing the sequala of surgery and radiotherapy. In this study, we promote a mpMRI-based prediction of csPCa (i.e., GG ≥ 3) by developing an ADC-based ML model. Indeed, the previous series showed the ADC prognostic value with adverse histologic patterns of PCa behaviors [8].

Our model shows very high performance in predicting GG ≥ 3, employing only four features. Furthermore, the signature highlights a strong similarity between GG1 and GG2 and significant differences between GG = [1,2] and GG ≥ 3. These findings seem to not support the current tendency to assume intermediate/high-risk PCa as belonging to the same class as csPCa [23]. Indeed, histologically, GG1 and GG2 can be considered more similar to each other than GG3 and GG4 are. A mild boundary between GG1 and GG2 exist, and, partly, it may be ascribed to their similar growth pattern, arising from the predominance of the “3” pattern, leading cancer glands to infiltrate normal tissue, while in the predominant “4” pattern (GG3 and GG4) and GG5, cancer cells tend to replace normal tissue [7]. Accordingly, GG1 and GG2 sometimes do not have a sharp separation even in the clinical pathway [24,25] since the pathway of GG2 is much more similar to that of low-risk PCa (GG1) rather than that of GG ≥ 3 [26]. Hence, despite a lack of consensus on this topic [27,28], this study highlights that patients with GG2 can still be considered non-csPCa; GG1 and GG2 (with lower metastatic risk) are identified as a single group in radiomic analysis. Therefore, as already happens for GG1, GG2 could even be suitable for active surveillance programs or novel focal ablative therapies [20,29]. Not least, this implies defining patients harboring csPCa as those with GG ≥ 3, which should be investigated for a potential metastatic burden and submitted to aggressive therapies, including radical treatments or combined approaches. Instead, from recent systematic reviews of the studies regarding mpMRI-based ML applications for PCa risk prediction, exploring the potentiality of PIRADS v2.1, the majority predicts as csPCa those lesions having GG > 1, and a small group refers to csPCa as those having GG > 4 [23,30]. Actually, only a few works [11,31,32] consider csPCa as GG ≥ 3. The study by Yual et al. [31] uses transfer learning to train a model consisting of unexplainable features on proprietary and public datasets. A direct comparison is prevented because only averaged measures are reported without any value distribution. As regards Cuocolo et al. [32], their study shows predictive performance lower than ours. Finally, the study by Ogbonnaya et al. [11] shows high predictive performance, using many more features and without performing any validation.

From a methodological point of view, our radiomic features yield parametric colormaps corresponding to ADC slices. This helps clinicians, in a multidisciplinary team, in feature interpretation and broader comprehension of tissue properties. In this regard, the radiomic signature highlights different properties of GG ≥ 3 compared to GG1 and GG2 lesions. First of all, μ−u shows that in GG ≥ 3, local mean ADC values are more uniform than in GG < 3. This considered, cv−m reports GG ≥ 3 as having a greater local variance in tissue-diffusive properties (i.e., ADC values); this also leads s−s to increase. At the same time, σ−IQR highlights that such a local variance stays more uniform in GG ≥ 3 than in GG < 3. Basically, given our fixed size unit (window) for local radiomic analysis in ADC slices, as cancer cells start growing at the early beginning, each local window has a low cancer cell density, which is a small within-unit variability of ADC values but is very different from unit to unit. As cancer grows, the within-unit variability of ADC values is expected to increase while the between-unit variability still remains. However, when the number of cancer cells grows to tend to occupy the whole unit, all the units are characterized by tissue with similar diffusivity properties (i.e., uniform local mean ADC values), and the between-unit variability tends to shrink accordingly. This finding is also in keeping with the intrinsic histological features of GG3 and GG > 3 tumors. GG3 tumors show intermingled normal glands by definition and, therefore, are a mixture of heterogeneous tissues. By contrast, GG > 3 tumors tend to build uniform aggregates of cancer glands, particularly in cribriform GG4.

Furthermore, the capability to implement an integration of conventional mpMRI evaluation with radiomic features by machine learning models could improve the discrimination between csPCa and non-csPCa, besides the re-assessment of mpMRI imaging by experienced radiologists. Moreover, the proposed parametric maps from ADC provide clinicians (primarily urologists and radiologists) with a useful tool that allows PCa lesions to be characterized even beyond the visible lesion boundaries. Besides that, additional advantages can be derived from multivariate hybrid models, including clinical parameters too, for csPCa patient stratification. The pre-biopsy evaluation of radiomic features by machine learning methods can help clinicians spare patients unnecessary biopsies, especially in borderline cases. For instance, patients with PIRADS 3 lesions can be jointly submitted to clinical and machine learning-based evaluation so that those with a high risk of csPCa by machine learning should be referred to target biopsy, whilst those with a low risk of csPCa could be counseled for follow up. At the same time, for those patients for which biopsy is required; this approach could even pave the way to radiomic-driven biopsy. Besides that, investigating the spatial correlation between the parametric feature maps and the digital scans of prostate histological samples can unveil cluster properties on the samples. In conclusion, with a wider population, mpMRI radiomic features can be integrated with clinic parameters and multi-modality features, including Prostate-Specific Membrane Antigen Positron Emission Tomography (PSMA-PET), to increase the precision of diagnosis of aggressive PCa needing to be treated.

Despite several strengths, our study is not devoid of limitations. First, the number of patients included in the study is limited; this possibly bounds the model’s generalizability, although that number allowed us to perform out-of-sample model validation. In addition, patients undergo mpMRI and MRI-target fusion biopsy in a single experienced center, thus increasing the homogeneity of the study population. Second, a ML model relying on bioptic samples may affect the accuracy in discriminating pathologic GG[1,2] from GG3 due to possible significant discrepancies between the clinic and pathologic GG. A further step is validating the model relying on pathologic specimens of patients who underwent radical prostatectomy.

## Figures and Tables

**Figure 1 cancers-14-06156-f001:**
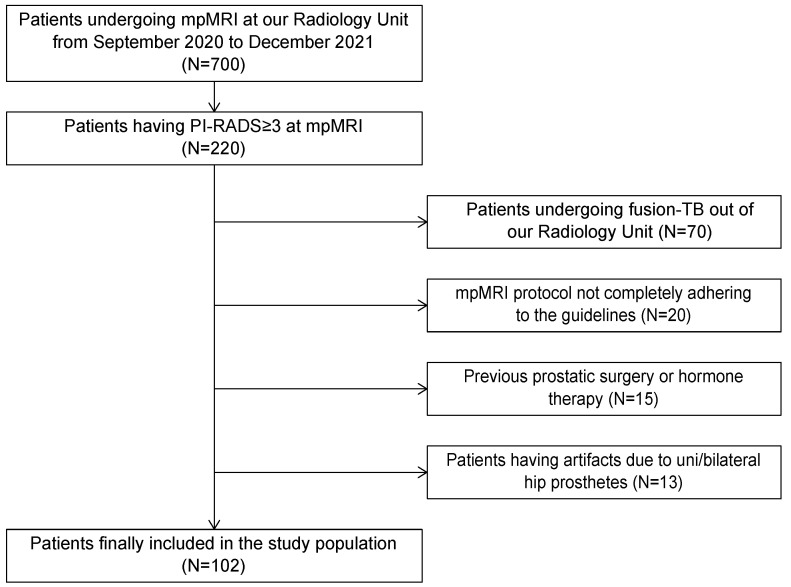
Flowchart of patient enrolment.

**Figure 2 cancers-14-06156-f002:**
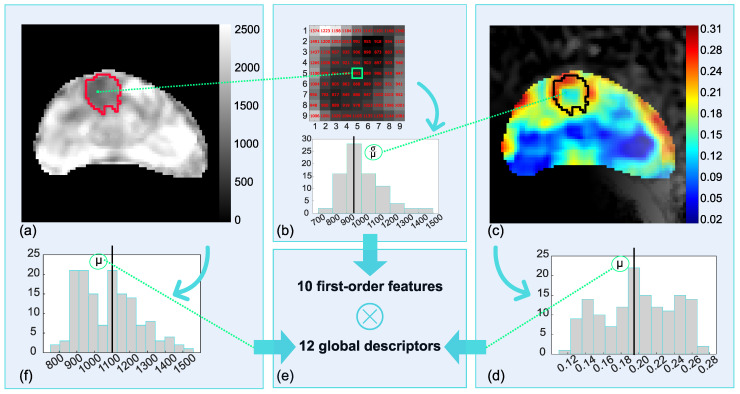
Generation of radiomic features using representative metrics. Altogether, 132 radiomic features are generated, stemming from the apparent diffusion coefficient (ADC) maps of the prostates (**a**), through exploiting the local approach for computing 10 first-order features in (**b**,**c**). Then, 12 global descriptors are computed on the histograms of the parametric maps (**d**), thus achieving 120 radiomic features (**e**). In addition, the same 12 global descriptors are computed directly on the ADC maps within the lesion region of interest (ROIs) (**f**).

**Figure 3 cancers-14-06156-f003:**
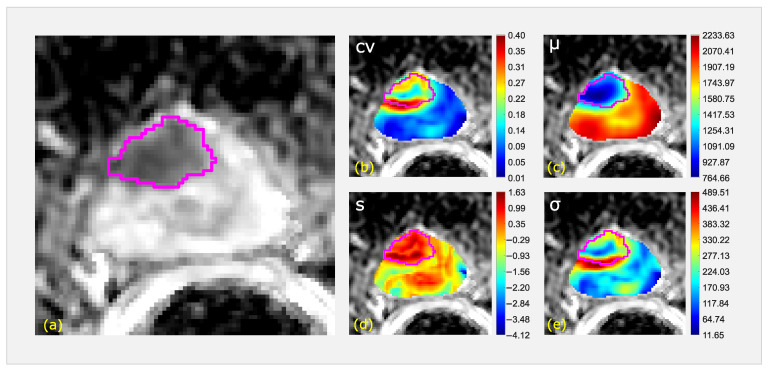
A representative ADC slice (**a**) for a non-csPCa (GG2) and the four colorimetric maps of cv (**b**), μ (**c**), *s* (**d**) and σ (**e**), with the lesion ROI superimposed in pink.

**Figure 4 cancers-14-06156-f004:**
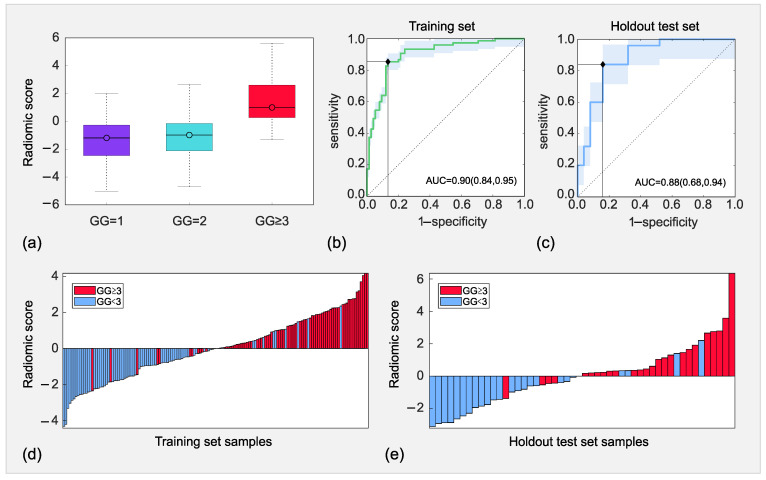
Boxplot of discrimination achieved by the radiomic score among the three Gleason Grade groups (GG)1, GG2, and GG ≥ 3 (**a**). receiver operating characteristic (ROC) curves achieved by the radiomic model in the training (**b**) and test (**c**) sets and waterfall plots achieved by the radiomic model in the training (**d**) and test (**e**) sets.

**Table 1 cancers-14-06156-t001:** Clinical characteristics of the patient cohort.

Parameter	Value
**Number of patients**	102
**Age**	
Mean ±σ (y)	71.3 ± 7.4
Range (y)	46.88
**PSA**	
Mean (ng/mL)	9.0
Range (ng/mL)	[1.6, 37.4]
**PSAD**	
<0.15 (ng/mL${}^{2}$)	52
≥0.15 (ng/mL${}^{2}$)	50

**Table 2 cancers-14-06156-t002:** Clinical characteristics of PCa lesions.

	GG < 3	GG ≥ 3
**Number of lesions**	61	56
**Lesion size**		
Mean (mm${}^{2}$)	78	119
Median (mm${}^{2}$)	60	69
IQR (mm${}^{2}$)	67	105
**GG**		
1	25	-
2	36	-
3	-	21
4	-	23
5		12
**PIRADS**		
3	34	9
4	22	35
5	5	12

**Table 3 cancers-14-06156-t003:** Performance of the radiomic model in the training and test sets.

Metric	Training (150 Samples)	Training (50 Samples)
**AUC**	0.90	0.88
**SN**	85%	84%
**SP**	87%	84%
**I**	0.72	0.68
**PPV**	87%	84%
**NPV**	85%	84%
**FP**	10	4
**FN**	11	4

## Data Availability

The data are not available because of patient privacy.

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
