# Peer review of "Beyond Multiparametric MRI and towards Radiomics to Detect Prostate Cancer: A Machine Learning Model to Predict Clinically Significant Lesions"

_cancers, 2022, doi:10.3390/cancers14246156_

Round 1

Reviewer 1 Report

Is an interesting paper about of develop a machine learning model to predict PCa. The paper is well written and the machine learning approach  is well conducted. The radiomic signature highlights shows differences compared with other studies.  The parametric maps from ADC could provide clinicians a useful tool to characterize PCA lesions even beyond the visible lesion. The discussion is well argued. I suggest a comparative Table  including the different approaches from other authors. Also  I suggest the authors could include on  the  discussion part , how the urologist  could use this ML in any part of the world and also the strengths and limitations.

Author Response

REVIEWER #1

General comments

[C1.0] Is an interesting paper about of develop a machine learning model to predict PCa. The paper is well written and the machine learning approach is well conducted. The radiomic signature highlights shows differences compared with other studies.  The parametric maps from ADC could provide clinicians a useful tool to characterize PCA lesions even beyond the visible lesion. The discussion is well argued.

[R1.0] We thank the reviewer for appreciating our study

Specific comments

[C1.1] I suggest a comparative Table including the different approaches from other authors.

[R1.1] What distinguishes our study from most of the existing ones is our proposal of using the definition of clinically significant PCa (csPCa) as Gleason Grade group (GG)≥3, instead of GG≥2, at biopsy examination. Therefore, would not have made sense comparing the results with studies having a different taxonomy. Accordingly, when discussing the other approaches, we have addressed only those three studies sharing our definition of csPCa, that is Yual at al., Cuocolo et al., Ogbonnaya et al. (Page 9, lines 242-247).

[C1.2] Also I suggest the authors could include on the discussion part, how the urologist could use this ML in any part of the world and also the strengths and limitations.

[R1.2] Thank you very much to prompt us to emphasize the benefits for urologists. To this purpose, we added the following part (Page 9, lines 267-281):

“Furthermore, the capability to implement an integration of conventional mpMRI evaluation with radiomic features by machine learning models could improve the discrimination between csPCa and non-csPCa, besides the re-assessment of mpMRI imaging by experienced radiologists. Not least, the proposed parametric maps from ADC provide clinicians (primarily urologists and radiologists) with a useful tool which allows characterizing PCa lesions even beyond the visible lesion boundaries. Besides that, additional advantages can derive from multivariate hybrid models, including clinical parameters too, for csPCa patient stratification. The pre-biopsy evaluation of radiomic features by machine learning methods can help clinicians to spare patients unnecessary biopsies, especially in borderline cases. For instance, patients with PIRADS 3 lesions can be jointly submitted to clinical and machine learning-based evaluation so that, those with high risk of csPCa by machine learning should be referred to target biopsy, whilst those with low risk of csPCa could be counselled for follow up. At the same time, for those patients for which biopsy is required, this approach […]”

As far as the strengths and limitations are concerned, they are already addressed in the first (Page 9, lines 248-267) and the last (Page 10, Lines 288-296) part of Discussion Section. Thus, to summarize, we identify two main limitations, one related to the limited size of the study population, the other referred to the use of biopsy outcome for model validation, prone to possible mismatch with respect to the histopathology grading.

Reviewer 2 Report

The manuscript is well written and clear, proposing clinically relevant results. Beautiful images.

Author Response

REVIEWER #2

General comments

[C2.0] The manuscript is well written and clear, proposing clinically relevant results. Beautiful images.

[R2.0] We thank the reviewer for her/his warm appreciation.

Reviewer 3 Report

Guadino et al. used a machine learning approach to classify the clinically significant Prostate Cancer (csPCa) by multiparametric Magnetic Resonance Imaging and compared them with the Gleason grading from 1 to 5. The study is very well designed and conceptualized. However, authors have not included adequate literature related to the topic. There are several studies which shows the correlation of PIRAD and aggressive PC at Radical prostatectomy (RP) pathology findings which must be included (Beksac et al. 2018, J. Urology; Hectors et al, 2019, J. Urol). As authors has pointed out the limitation of their studies that it lacks the pathological finding at RP. It will be better to include the advantages of their studies over the previously reported studies which had already shown the utility of mpMRI in detecting the aggressive form of cancers.

Beksac AT et al. Multiparametric Magnetic Resonance Imaging Features Identify Aggressive Prostate Cancer at the Phenotypic and Transcriptomic Level. J Urol. 2018 Dec;200(6):1241-1249. doi: 10.1016/j.juro.2018.06.041. Epub 2018 Jul 3. PMID: 30563651.

Hectors SJ et al. Radiomics Features Measured with Multiparametric Magnetic Resonance Imaging Predict Prostate Cancer Aggressiveness. J Urol. 2019 Sep;202(3):498-505. doi: 10.1097/JU.0000000000000272.

Author Response

REVIEWER #3

Specific comments

[C3.0] Guadino et al. used a machine learning approach to classify the clinically significant Prostate Cancer (csPCa) by multiparametric Magnetic Resonance Imaging and compared them with the Gleason grading from 1 to 5. The study is very well designed and conceptualized. However, authors have not included adequate literature related to the topic. There are several studies which shows the correlation of PIRAD and aggressive PC at Radical prostatectomy (RP) pathology findings which must be included (Beksac et al. 2018, J. Urology; Hectors et al, 2019, J. Urol).

Beksac AT et al. Multiparametric Magnetic Resonance Imaging Features Identify Aggressive Prostate Cancer at the Phenotypic and Transcriptomic Level. J Urol. 2018 Dec;200(6):1241-1249. doi: 10.1016/j.juro.2018.06.041. Epub 2018 Jul 3. PMID: 30563651.

Hectors SJ et al. Radiomics Features Measured with Multiparametric Magnetic Resonance Imaging Predict Prostate Cancer Aggressiveness. J Urol. 2019 Sep;202(3):498-505. doi: 10.1097/JU.0000000000000272.

[R3.0] We thank the reviewer for her/his appreciation and for comments and suggestions. We agree that a conspicuous literature on predicting csPCa exists, as we have indirectly highlighted citing the two systematic reviews in [23, 30]). The problem is that we propose not to use the most common definition of csPCa, that is Gleason Grade group (GG)≥2, therefore we believed that was more appropriate and interesting only addressing those papers which shared with us the same definition of csPCa, that is (GG)≥3 at biopsy examination (i.e., Yual at al., Cuocolo et al., Ogbonnaya et al.). This notwithstanding, we are aware that some discrepancies may arise from biopsy and pathology findings from radical prostatectomy, as we report in limitations (Page 9, Lines 282-286):

“[…] a ML model relying on bioptic samples may affect the accuracy in discriminating pathologic GG[1,2] from GG3, due to possible significant discrepancy between clinic and pathologic GG. A further step is validating the model relying on pathologic specimens of patients who underwent radical prostatectomy.”

Therefore, we chose to extend literature discussion to those studies based on pathology grading once even our predictive model will be validated on pathology grading.

[C3.1] As authors has pointed out the limitation of their studies that it lacks the pathological finding at RP. It will be better to include the advantages of their studies over the previously reported studies which had already shown the utility of mpMRI in detecting the aggressive form of cancers.

[R3.1] We thank the reviewer for this comment, so we had the opportunity to clarify this point in the manuscript too, by rephrasing the following sentence (Page 9, lines 232-239):

“Hence, despite a lack of consensus on this topic [27,28], marking a breaking point with previous works, this study highlights that patients with GG2 can still be considered as non-csPCa, being GG1 and GG2 (with lower metastatic risk) identified as a single group at radiomic analysis. Therefore, as already happens for GG1, GG2 could even be suitable for active surveillance program or novel focal ablative therapies [20,29]. Not least, this implies defining patients harbouring csPCa as those with GG≥3, which should be investigated for potential metastatic burden and submitted to aggressive therapies including radical treatments or combined approaches. Instead, from recent systematic reviews of the studies […]”

Indeed, although using a validation based on fusion biopsy examination, the major advantage highlighted by our study is that PCa graded as GG=2 can still be considered as ncsPCa, so possibly admitted to Active Surveillance protocols instead of undergoing a radical treatment. The confirmation coming from pathological findings at RP will strengthen this result, providing it with a marked clinical impact.

Reviewer 4 Report

1) The contributions of the proposed work are not included at the end of the introduction

2) The performances of the proposed method can be compared with at least three existing methods for prostate cancer detection

3) Discuss the limitations of the existing approaches for prostate cancer detection and how the proposed approach overcomes these limitations

4) Discuss the limitations of the proposed approach.

5) Discuss future works

6) Authors can add some visualization to show the features

7) What is the motivation behind choosing SVM and why not other machine learning classifiers?

8) The parameters of SVM not studied in detail? Why? In addition, the authors did not include the values for all the parameters. This is important and without this, the experiments cannot be reproducible by others

Author Response

REVIEWER #4

Specific comments

[C4.0] The contributions of the proposed work are not included at the end of the introduction

[R4.0] Thank you for your comment. We have rephrased the last sentence of the Introduction to state explicitly the contributions of our study, as follows (Page 2, Lines 51-56):

“Accordingly, the main goal of our research is to develop a ML model to predict csPCa namely, GG≥3. In particular, this study sets up an image processing procedure based on extracting local radiomic features, enabling the visual assessment of colorimetric maps on the whole prostate gland, through which analysing similarities and differences between GG1, GG2, and GG≥3, besides making prediction on high-risk diseases, even before any biopsy examination.”

[C4.1] The performances of the proposed method can be compared with at least three existing methods for prostate cancer detection

[R4.1] We did it. In fact, we addressed the critical discussion of the existing literature, by including those works with which a fair comparison with our study was feasible, based on clinical hypothesis and predictive performance, as indicated at Page 9, lines 238-247:

“From recent systematic reviews of the studies regarding mpMRI-based ML applications for PCa risk prediction, exploring the potentiality of the PIRADS v2.1, the majority predicts as csPCa those lesions having GG>1, and a small group refers to csPCa as those having GG>4 [23,30]. Actually, just a few works [11,31,32] consider csPCa as GG≥3. The study by Yual et al.[31] uses transfer learning to train a model consisting of unexplainable features on proprietary and public datasets. A direct comparison is prevented because only averaged measures are reported, without any value distribution. As regards Cuocolo et al.[32], their study shows predictive performance lower than ours. Finally, the study by Ogbonnaya et al.[11] shows high predictive performance, using many more features and without performing any validation.”

[C4.2] Discuss the limitations of the existing approaches for prostate cancer detection and how the proposed approach overcomes these limitations

[R4.2] In our opinion, the main limitation of the majority of the existing approaches is represented by the assumption of considering clinically significant Prostate Cancers (csPCa) as those having Gleason Grade Group (GG)≥2, instead of GG ≥3, as generally considered by the majority of existing studies. In this respect, what written  at Page 8-9, Lines 222-231, represents a discussion of the present limitations and a proposal to overcome them:

“[…] These findings seem not supporting the current tendency to assume intermediate/high-risk PCa as belonging to the same class as csPCa [23]. Indeed, histologically, GG1 and GG2 can be considered as being more similar between each other, than GG3 and GG4 are. A mild boundary between GG1 and GG2 exist and, partly, it may be ascribed to their similar growth pattern, arising from the predominance of the “3” pattern, leading cancer glands to infiltrate within normal tissue, while in predominant “4” pattern (GG3 and GG4) and GG5, cancer cells tend to replace normal tissue [7]. Accordingly, GG1 and GG2 sometimes have not a sharp separation even in the clinical pathway [24,25], since the pathway of GG2 is much more similar to that of low-risk PCa (GG1) rather than that of GG≥3 [26].”

Moreover, our approach is characterized by explainable local radiomic features, endowed with colorimetric maps, that representing a novel approach to help researcher in  their clinical interpretation when superimposed on prostate gland morphology, this representing a novel approach to help researcher to assess by themselves whether the abovementioned limitation can be overcome (Page 8, Lines 248-252):

“From a methodological point of view, our radiomic features yield parametric colormaps corresponding to ADC slices. This helps clinicians, in a multidisciplinary team, in feature interpretation and a broader comprehension of tissue properties. In this regard, the radiomic signature highlights different properties of GG≥3 comparing to GG1 and GG2 lesions”. On the other hand, feature interpretation is also favoured by the limited number of radiomic features included in the predictive model, and this seems to be very hard to be found in previous studies, like Yuan et al. or Ogbonnaya et al., which use many more unexplainable features.”

[C4.3] Discuss the limitations of the proposed approach.

[R4.3] Actually, this is what we did at the end of Discussion section (Page 9-10, Lines 278-286):

“Despite several strengths, our study is not devoid from limitations. First, the number of patients included in the study is limited, this possibly bounding the model generalizability, although that number allowed us to perform an out-of-sample model validation. In addition, patients undergo mpMRI and MRI-target fusion biopsy in a single experienced Centre, thus increasing the homogeneity of the study population. Second, a ML model relying on bioptic samples may affect the accuracy in discriminating pathologic GG[1,2] from GG3, due to possible significant discrepancy between clinic and pathologic GG. A further step is validating the model relying on pathologic specimens of patients who underwent radical prostatectomy.”

Thus, to summarize, we identify two main limitations, one related to the limited size of the study population, the other referred to the use of biopsy outcome for model validation, prone to possible mismatch with respect to the histopathology grading.

[C4.4] Discuss future works

[R4.4] We did include insights on future works in the last lines of the Discussion section, by jointing their discussion with conclusive remarks, here reported for the sake of clarity:

  • [Page 9, Lines 274-277] “[…] with a wider population mpMRI radiomic features can be integrated with clinic parameters and multi-modality features, including Prostate Specific Membrane Antigen Positron Emission Tomography (PSMA-PET), to increase the precision of diagnosis of aggressive PCa needing to be treated.”
  • [Page 10, Lines 284-286] “A further step is validating the model relying on pathologic specimens of patients who underwent radical prostatectomy.”

To summarize, future works will address, firstly, the development of hybrid predictive models for csPCa, combining radiomic features with clinic parameters and PSMA-PET-derived features and, secondarily, the validation of the predictive model with histopathology grading.

[C4.5] Authors can add some visualization to show the features 

[R4.5] This is really a valuable suggestion, thank you very much. We added Figure 3 to show some colorimetric maps of local features on the prostate gland, which is introduced in Section 3.2, as follows:

“Figure 3 shows an example, for a representative ADC slice (a) of a ncsPCa (GG2), of the four colorimetric maps of cv (b), µ (c), s (d), and σ (e).”

[C4.6] What is the motivation behind choosing SVM and why not other machine learning classifiers?

[R4.6] Thank you very much for your question. We self-censored some technical details and explanations of methodological choices thinking at the present journal, Cancers, as not being specifically dedicated to technical and engineering readers, so that your question gave us the possibility to include this information and enrich the quality of the manuscript. Accordingly, we include the following sentence to justify the choice of the SVM for classifier development (Page 5, Lines 152-158):

“A linear SVM classifier was adopted for predictive model development because it can work well even with reduced sample datasets, being based on Support Vectors (SVs) placement within the feature space. In addition, the linear kernel uses the minimum number of hyperparameters, thus contributing to minimize the risk of overfitting on the original population. Finally, this choice allowed us to perform a fair comparison of the models developed on the original population and the augmented ones.”

[C4.7] The parameters of SVM not studied in detail? Why? In addition, the authors did not include the values for all the parameters. This is important and without this, the experiments cannot be reproducible by others

[R4.7] For the reasons explained in [R4.6] we avoided to include some technical details, that now have finally found home in the manuscript, in M&M and Results sections. In particular, the following sentences have been added:

  • [Page 5, Lines 162-164] […]and for each run, the SVM hyperparameters, linear scale and misclassification cost C, have been estimated using the built-in MatLab Bayesian optimization algorithm

The reference number 19 has been added accordingly.

  • [Page 8, Line 198] The developed SVM predictive model had =0.0366$ and C=0.0010.